# Continuous-Flow Microwave Milk Sterilisation System Based on a Coaxial Slot Radiator

**DOI:** 10.3390/foods12030459

**Published:** 2023-01-18

**Authors:** Junhui Guo, Huacheng Zhu, Yang Yang, Qinggong Guo

**Affiliations:** College of Electronic and Information Engineering, Sichuan University, Chengdu 610065, China

**Keywords:** microwave heating, continuous-flow, milk sterilization, heating efficiency, heating uniformity

## Abstract

Microwave continuous-flow liquid food sterilisation, in which the liquid is mainly heated by microwaves, has the advantages of fast sterilisation speed, energy saving, comprehensive elimination, and less nutrient loss. Circular pipes are commonly used in microwave continuous-flow liquid heating processing. However, with circular pipes, which are widely used in the industry, the heating is uneven owing to the phenomenon of tube focusing when adopting external radiation. In this study, a novel microwave continuous-flow milk sterilisation system based on a coaxial slot radiator is proposed. First, the coaxial slot radiator was designed to realise efficient radiation through the establishment of multi-physics model. The structure of the system was then optimised by comparing the heating efficiency and uniformity of simulation results. The effect of microwave coaxial slot radiator rotation on heating uniformity was simulated and the results show that the heating uniformity is improved obviously. Experimental equipment was set up to verify the results of the simulation. The experimental results are consistent with the simulation results. Finally, the sensitivity analysis of the system is performed to confirm that, when the dielectric properties and types of liquid food change, the heating of the proposed microwave continuous-flow system remains efficient and uniform.

## 1. Introduction

Microwave energy has been widely used as an efficient, and clean energy in many fields, such as material extraction, thermal engineering, and chemical reactions, [1,2,3,4]. Given that microwave food processing can reduce the consumption of nutrient substances caused by the process and to ensure food quality [5,6], microwave technology has been extensively studied for applications in the food processing industry, such as food sterilisation, vacuum drying, and liquid food processing [7,8,9,10,11,12]. Compared with the traditional liquid food sterilisation technology based on pasteurised and high-temperature instantaneous sterilisation, microwave liquid sterilization, with the advantages of high heating efficiency and uniformity, low nutrition loss, sustainability, and reduced time and cost, is expected to have broad application prospects in liquid sterilization systems [5,13,14,15,16,17,18]. To meet the requirement for continuous processing of liquid food, microwave continuous-flow treatment is a feasible method that translates batch processing into continuous processing [19,20,21] and has been widely used in organic synthesis engineering, extraction engineering, etc. [22,23,24]. However, the non-uniform temperature distribution induced by non-uniform distribution of electromagnetic fields and the dielectric properties, viscosity, and the flow rate of liquid severely restricts the wide application of the microwave continuous-flow system [13,25,26,27]. To realise microwave continuous-flow liquid processing systems applying a wider range of application scenarios, more uniform microwave continuous-flow systems are required. Numerous studies have been conducted by many scholars and experts. Tuta and Palazoglu designed a novel continuous-flow microwave heating unit in which high-viscosity liquid food products flow in a helical tube. The effect of liquid flow velocity and liquid viscosity was studied. The numerical and experimental results showed that more uniform heating can be achieved for high-viscosity fluid food products by using a helical tube in continuous-flow microwave heating systems [28]. Topcam et al. developed a mathematical model to enable the liquid to rotate to strengthen the mixing of materials and validated that liquid rotation can greatly improve the temperature uniformity of a low-high viscous liquid through simulations and experiments [29]. Ye et al. established a microwave continuous-flow heating system model with a screw propeller in the material tube to study the effect of the spiral propeller rotation on heating uniformity. The results indicated that the heating uniformity was increased by 90% approximately by rotating screw propeller and the larger the speed of the spiral propeller, the better the heating uniformity [30]. Zhang et al. simulated a spiral tube microwave continuous-flow heating system to analyse the effect of the structural parameters of the tubes on heating efficiency and uniformity and concluded that the longer the pitch of the spiral tube, the better the heating uniformity [31]. However, most of the microwave continuous-flow heating systems have large resonant cavities, which lead to high manufacturing cost and difficulty for scaling up [13,32,33] and use external radiation that is not adapted for circular pipes, which are widely used in the continuous-flow field due to the occurrence of tube focusing resulting in non-uniform heating. Coaxial radiator, which has the scalability that is hardly attainable in rectangular or circular waveguides widely used in microwave technology and realises the internal radiation, has broad application prospects in the field of microwave continuous-flow heating [34]. Abdullah designed a coaxial slot antenna applicator that achieves efficient and uniform sterilisation of the stationary milk. The temperature uniformity of this applicator was 89.2% better than the uniformity of previous milk microwave batch pasteurisation and the pasteurisation efficiency according to the APC test was 99.99% [35]. The microwave milk sterilisation system above based on the coaxial slot antenna was designed for stationary milk, however research on the microwave continuous-flow sterilization based on the coaxial slot radiator are insufficient.

In this study, a novel microwave continuous-flow milk sterilisation system based on a coaxial slot radiator was developed to realise high heating efficiency and uniformity for milk continuous-flow sterilization while its structure is simple. In Section 2, how COMSOL was used to establish a multi-physics model of this microwave continuous-flow system is described. In Section 3, dimension optimisation of the coaxial slot radiator and the microwave continuous-flow milk sterilisation system were carried out, respectively, and the effect of the coaxial slot radiator rotation on heating uniformity was studied. Experimental studies were conducted to validate simulation results. Finally, the sensitivity analysis of the microwave milk sterilisation system was performed.

## 2. Multiphysics Simulation

### 2.1. Geometry

The 3D geometry of the experimental system was developed in the commercial finite element software COMSOL Multi-physics (6.0a, COMSOL Inc., Stockholm, Sweden). The model consists of a microwave coaxial slot waveguide, hollow-ring quartz tube, and hollow-ring stainless steel container, as shown in Figure 1. A microwave coaxial-slot radiator consists of an inner conductor and outer conductor. The inner conductor is a cylindrical aluminium shaft with a radius of 4 mm and a length of 318 mm, whereas the outer conductor is a hollow-ring aluminium tube with an inner diameter of 19 mm, outer diameter of 25 mm, and length of 320 mm. There are four slots with a depth of 11.5 mm and width of 5 mm around outer conductor. A hollow-ring quartz tube with an outer diameter of 29 mm and inner diameter of 26 mm is enclosed out of the microwave coaxial slot radiator, outside of which there is a hollow-ring stainless steel container, with an inner diameter of 35 mm and outer diameter of 41 mm.

### 2.2. Governing Equations

The model simulation was coupled with an electromagnetic field, heat transfer, and liquid flow field [34]. Maxwell’s equations were used to calculate the electromagnetic field:(1)$$\begin{array}{c} \nabla \times H(r,t)=J+\varepsilon \frac{\partial E}{\partial t}\\ \nabla \times E(r,t)=-\frac{\partial B}{\partial t}\\ \nabla \cdot B(r,t)=0\\ \nabla \cdot D(r,t)=\rho_{e} \end{array}$$
where ***H*** and ***E*** are the magnetic and electric field intensities, respectively; **r** represents the coordinate radius vector; *t* represents time; ***J*** is the ampere density; ***B*** is the magnetic induction intensity; ***D*** is the electric displacement vector; *ρ*_e_ is the electric charge density; and $\nabla \equiv e_{x}\frac{\partial }{\partial x}+e_{y}\frac{\partial }{\partial y}+e_{z}\frac{\partial }{\partial z}$ is the vector differential operator. The wave equation of the electric field can thus be derived from Equation (1) and expressed as the Helmholtz equation:(2)$$\begin{array}{c} \nabla \times \mu_{r}^{-1}(\nabla \times E)-k^{’}{}^{2}(\varepsilon_{r}\varepsilon_{0}-\frac{j\sigma }{\omega })E=0\\ k^{’}=\omega \sqrt{\varepsilon_{0}\mu_{0}} \end{array}$$
where *μ_r_* and *k^′^* denote the relative permeability, and wave number in free space, respectively; ***ε****_r_* is the relative permittivity; ***ω*** is the angular frequency; ***σ*** is the electrical conductivity; and ***ε***_0_ and *μ*_0_ represent the permittivity and permeability of the vacuum, respectively.

The electromagnetic power loss *Q_e0_* can be derived from Equation (2) by applying the electric-field strength and dielectric properties:(3)$$Q_{e0}=\frac{1}{2}\omega \varepsilon_{0}\varepsilon_{r’}{\left|E\right|}^{2}$$
where ***ε****_r′_* is the imaginary part of the relative permittivity of the processing material. Supplementing the equations above with the Fourier equation and fluid-flow relationship, the heat transfer equation can be derived to calculate the distribution temperature of the heated object, which can be expressed as follows:(4)$$\rho_{0}C_{p}(\frac{\partial T}{\partial t}+u\cdot \nabla T)=\nabla (\kappa \nabla T)+P_{d}$$
where *ρ*_0_ is the liquid density; *C_p_* is the material heat capacity; *T* is the temperature of the liquid; ***u*** is the velocity of the liquid; κ is the thermal conductivity of the material; and *P_d_* is the electromagnetic power density.

The velocity distribution of the flowing liquid can be obtained by solving the continuity equation, which expresses the conservation of mass: (5)$$\frac{\partial \rho }{\partial t}+\nabla \cdot (\rho u)=0$$
and the Navier-Stokes equation, which expresses the conservation of momentum: (6)$$\rho (\frac{\partial u}{\partial t}+u\cdot \nabla u)=-\nabla P+\mu \nabla^{2}u+\rho g$$
where **▽***P* is the pressure intensity per unit volume; ***u*** and **g** represent the viscosity and gravitational acceleration, respectively; *μ* represents the constant viscosity of the liquid; and *μ***▽**^2^***u*** is equal to **▽***T*, which is the divergence of the stress deviator tensor.

### 2.3. Input Parameters

In this model, the frequency of the electromagnetic wave was set to 2.45 GHz. This is the free-licence ISM frequency defined by the ITU that is commonly used in the food industry. The input power of the electromagnetic wave to the coaxial slot waveguide was set to 1 kW. It was also assumed that the initial temperature of the milk in the cavity is 293.15 K. Related input parameters of the simulation are listed in Table 1 and the dielectric properties of the milk at different temperatures at 2.45 GHz are presented in Table 2 [35,36].

## 3. Results and Discussion

### 3.1. Model Validation

To verify the feasibility of the simulation model, the above-mentioned continuous-flow sterilisation system was set up for continuous-flow heating experiments. The microwave continuous-flow system and the coaxial slot radiator are shown in Figure 2 and the experimental setup for model validation is presented in Figure 3. This microwave continuous-flow system was used for milk sterilisation; continuous milk flows into the inlet and flows out of the outlet, and the pump guarantees that the rate of milk entering the inlet is constant. The microwave solid-state generator, circulator, and water load are manufactured by Chengdu Mapping Power Technology Company Ltd., Chengdu, China. They were used to generate microwaves and ensure safe microwave transmission. A directional coupler (Loop-E22DC40A10N, Euler Microwave, Sichuan Province, Chengdu, China) and microwave power meter (AV2433, 41st Institute of China Electronics Technology Group Corporation, Anhui Province, China) were used to measure the microwave power. The energy-radiation-efficiency measuring device comprised a microwave network analyser (E83363c, Agilent Technologies, Inc., Wood Dale, IL, USA), dielectric probe kit (N1501A, Agilent Technologies, Inc., Wood Dale, IL, USA), test software, and thermoelectric couple (TP678, MITIR, Zhejiang Province, Wenzhou, China).

The microwave frequency was set to 2.45 GHz. Full-fat milk (Mengniu Dairy Co., Ltd., Tangshan, China) was heated. Parameter S11 of the continuous-flow system with different variation of coaxial slot radiator rotation angles were measured using a vector network. The results are shown in Figure 4.

The simulation value of S11 was −12.5 dB. It agrees with the experimental results within an accuracy of 2%. Therefore, the simulation results can be assumed acceptable given that the disturbances caused by the coaxial transitions, openings, and soldering peculiarities were ignored in the simulation.

The point temperature of the outlet surface was also measured using thermocouples to verify the accuracy of the simulation results. The initial temperature of milk was 294.8 K and the rate of milk flow into the inlet was 10 mm/s. The microwave frequency was set to 2.45 GHz; the input power was set to 100 W and the heating time was 50 s. A comparison of the point temperature of the outlet surface from the simulation and from experiments as a function of time is shown in Figure 5. It can be concluded that the experimental and simulation results generally agree well but are not completely consistent. A possible reason may be that the existence of partial heat loss was not considered in the simulation.

### 3.2. Design of Coaxial Slot Radiator

In this section, the effects of slot angle and slot depth on the radiation efficiency of the coaxial slot radiator are discussed. The radiation efficiency of the coaxial slot radiator in air was calculated by simulation. In this model, the electromagnetic wave frequency was set to 2.45 GHz and the electromagnetic power input to the coaxial slot radiator was set to 1 kW. There was air outside the coaxial slot radiator. It is well-known that the energy utilisation efficiency can be expressed by the parameter S11, which represents the return energy loss of the microwave radiator. The smaller the value of S11, the more efficient the microwave radiator. The design of the coaxial slot radiator structure discussed in this section includes slot angle and slot depth. The angle of the slot ranged from 36° to 165.6°, with a step size of 14.4°and the slot depth ranged from 5 mm to 30 mm, with a step size of 1 mm. The values of S11 of the microwave coaxial slot radiator varying in slot depth and angles are shown in Figure 6. It can be observed that, when the slot angle is greater than 136.8°, the efficiency of the coaxial slot radiators is higher. When the slot depth is between 5 and 10 and the slot angle is greater than 136.8°, S11 is less than −7 dB, which means that these models have a decent energy efficiency.

Considering the impact of the slit angle on the heating uniformity, coaxial slot radiators with greater slot angles are more suitable. Therefore, a coaxial slot antenna with a slot depth of 5 mm and slot angle of 165.6° is assumed in the subsequent calculations.

### 3.3. Dimensional Optimisation of Heating System

After determining the dimensions of the coaxial slot radiator, the structure of the microwave system outside the coaxial slot radiator is discussed. The dimension optimisation of the microwave system mainly considers two factors: Radiating efficiency and field uniformity. The radiation efficiency is expressed by the parameter S11. A microwave system can be applied in engineering when the value of S11 of the microwave system is less than −10 dB. It is also necessary to study the uniformity of radiant energy, which can be measured by the covariance (COV) of the heating field temperature. We used the calculation method of COV reported by Zhu et al., 2018 [3]:(7)$$\mathrm{COV}=\sqrt{\frac{\sum_{i=1}^{n}\left(T_{i}-T_{a}\right)}{n}}/(T_{a}-T_{0})$$
where *T_i_* represents the point temperature of the selected region, *n* is the total number of points, and *T_a_* and *T*_0_ represent the average temperature and initial average temperature, respectively. Note that the smaller the value of COV, the greater the uniformity.

Given that the designed coaxial slot radiator was inserted into the quartz ring tube, to achieve high radiating efficiency and uniformity, the key parameters are the quartz-tube thickness, distance between the coaxial radiator and quartz tube, and thickness of the milk-container ring tube. Models with different quartz-tube thicknesses, distances between the coaxial slot radiator and quartz tube, and thicknesses of the milk-container ring tube were simulated using COMSOL. The thickness of the milk-container ring tube ranged from 3.5 mm to 4.5 mm with a step size of 0.5 mm. The quartz-tube thickness ranged from 0.5 mm to 2 mm, and the distance between the coaxial slot radiator and quartz tube ranged from 0.5 mm to 3 mm, with a same step size of 0.25 mm. The operating frequency was 2.45 GHz and the input electromagnetic wave power was 1 kW. In the simulation analysing the S11 of the microwave system, the milk was 293.15 K in temperature, 64.39 in dielectric constant and 13.68 in loss factor [37]. In the simulation analysing heating uniformity of the microwave system, the initial temperature of the milk was 293.15 K, heating time was 100 s, and the dielectric property of the milk was dependent on the temperature. The S11 and COV simulation results of the models with different size parameters are shown in Figure 7 and Figure 8, respectively. It can be concluded from the simulation results that, when the thickness of the milk-container ring was 3 mm or 4 mm, most of the models met the application requirements with S11 less than −10 dB. When the thickness of the milk-container ring was 3.5 mm; the quartz-tube thickness was smaller than 1.2 mm and the distance between the coaxial radiator and quartz tube was less than 0.8 mm, the value of S11 from the model was smaller than −10 dB. Nevertheless, the heating uniformity was better when the thickness of the milk-container ring was 3 mm. Consequently, a microwave system with a quartz-tube thickness of 1.5 mm, a distance between the coaxial radiator and quartz tube of 0.5 mm, and a thickness of the milk-container ring tube of 3.5 mm is preferable. This was the system chosen for further study.

### 3.4. Analysis of Rotating Radiator Effect

The target of high efficiency of the proposed microwave continuous-flow sterilisation system was achieved after dimensional optimisation. Further study on the effect of enabling the coaxial slot radiator to rotate on the heating uniformity was conducted. In this section, the effect of liquid continuous flow on heating uniformity was also considered in the simulation. The effect of different rotation angles of the coaxial slot radiator per second on the uniformity was studied. The frequency was set to 2.45 GHz and the input power was 1 kW. The initial temperature of the milk was 293.75 K and inflow velocity was 12 mm/s. The simulated operating time was 30 s. The angle variations per second of the coaxial slot radiator rotation ranged from 10° to 90°. The simulation results are shown in Table 3 and Figure 9. Comparing the COV results of the model with and without coaxial slot radiator rotation in Table 3, it can be seen from the simulation results that heating uniformity can be effectively enhanced through rotating the coaxial slot radiator. Further, when the rotation angle per second is 50°, the heating uniformity is relatively better.

### 3.5. Sensitivity Analysis

#### 3.5.1. Effect of the Bandwidth

The operating frequency and relative permittivity are both important parameters that affect heating efficiency and uniformity. The applicability of the microwave continuous-flow sterilisation system to different operating frequencies and milk with different dielectrics were simulated. In this section, considering that the microwaves generated by the microwave sources are not guaranteed to be constant at 2.45 GHz in frequency, the heating efficiency of this system at different operating frequencies are simulated. The operating frequency ranged from 2.35 GHz to 2.5 GHz. The system input power was 1 kW, whereas the heating time was set to 100 s. milk, and the inlet flow rate of the continuous liquid entering the pipe was set to 3.2 mm/s. The simulation results are shown in Figure 10. when the operating frequency varies, the S11 values of this microwave system exhibit a slight change, but remain lower than −10 dB, which confirm that the proposed microwave continuous-flow system can maintain high radiant efficiency in practical application.

#### 3.5.2. Effect of the Dielectric Constant of the Liquid

Next, heating simulation of milk with different dielectric constants was performed. The operating frequency was set to 2.45 GHz and the input power was set to 1 kW. The initial temperature of the liquid was 293.15 K and the rate of continuous flow into the pipe was 3.2 mm/s. The loss factor of the liquid was fixed as 13.68 and the heating time was 100 s. Dielectric constants of the liquid ranged from 58 to 70. The simulation results are shown in Figure 11. It can be observed that during the gradual increase of the dielectric constant, the value of S11 decreases and the value of COV increases. Figure 11 proves that the values of S11 are all less than −12 dB and COV are all less than 0.0275 with different dielectric constants. The simulation results indicate that for liquids with different real parts of relative permittivity, heating processes of the proposed microwave continuous-flow system is efficient and uniform.

#### 3.5.3. Effect of the Dielectric Loss Tangent of the Liquid

Further study on the effect of liquid with different dielectric loss tangents under a fixed dielectric constant was performed. The dielectric loss tangents of the liquid varied from 0.15 to 0.3 and the dielectric constant was 64.39. The initial temperature of the liquid was 293.15 K and the liquid flow was at an inflow velocity of 3.2 mm/s. The microwave frequency was 2450 MHz and the input power was set as 1000 W. The heating time was 100 s. The simulation results of S11 and COV are shown in Figure 12. It can be observed that although the dielectric constant changed, the values of S11 remained lower than −10 dB and COV remained less than 0.027. It demonstrates that the proposed microwave continuous-flow system maintains high heating efficiency and uniformity for liquid with different dielectric loss tangents.

#### 3.5.4. Effect of Various Liquid Food on Heating Performance

The heating effect of the proposed microwave system on different liquid foods was studied. The heating process of various liquid foods, including grape juice, apple juice, pear juice, orange juice, and pineapple juice were simulated. The dielectric properties of five liquid foods versus temperature at 2.45 GHz are presented in Table 4 [38].

The heating simulation ran for 100 s with an input power of 1 kW at a frequency of 2.45 GHz. The initial temperature of the liquid food was 293.15 K and the inflow velocity was 0.01 m/s. The simulation results for energy efficiency and heating uniformity are shown in Figure 13. Considering that the energy efficiency was always greater than 90% and the COV values were all less than 0.12, this microwave system can be assumed to be applicable for the sterilisation of different liquid foods.

## 4. Conclusions

In this study, a novel microwave continuous-flow milk sterilisation system based on a coaxial slot radiator was developed. It realises efficient and uniform heating of continuous-flow liquid during the milk sterilisation process. The FEM algorithm was utilised to design a coaxial slot radiator and optimised the entire structure of the microwave continuous-flow system. The effect of rotating the coaxial slot radiator on heating uniformity was performed and the simulation results show that the heating uniformity is greatly improved. The COV of the outlet surface was reduced from 0.31 to 0.13. An experimental system was established to validate the multiphysics model. The experimental and simulation results agreed well. A sensitivity analysis was performed to confirm that the heating effect on liquid foods with different dielectric properties was efficient and uniform. The simulation results demonstrate that the microwave absorption rate was above 90% in most cases. In addition, this microwave continuous-flow milk sterilisation system can be useful in other continuous-flow liquid treatment environments far beyond the milk sterilisation field. In summary, compared with other continuous-flow liquid treatment devices, the proposed microwave continuous-flow system is relatively simple, achieves continuous-flow liquid processing with high efficiency and uniformity, and is easy to scale up. However, considering that the proposed microwave continuous-flow system is small lab scale, it is worth studying the heating efficiency and heating uniformity when the system scales up. This work will be of substantial benefit for the design of microwave continuous-flow system production and scaled-up technologies.

## Figures and Tables

**Figure 1 foods-12-00459-f001:**
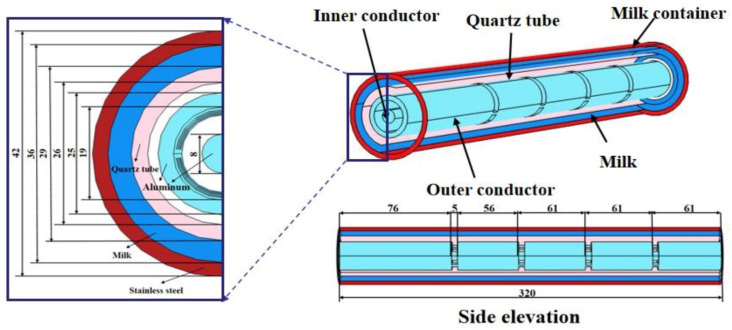
The geometry of the simulation model (unit: mm).

**Figure 2 foods-12-00459-f002:**
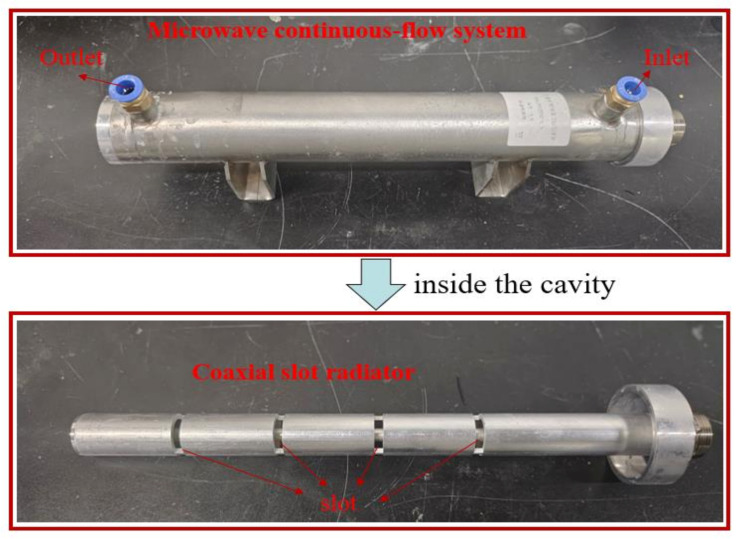
Microwave continuous-flow milk sterilisation system.

**Figure 3 foods-12-00459-f003:**
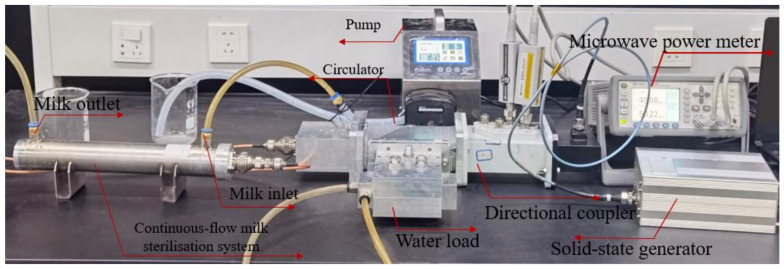
Experimental setup of the proposed continuous-flow milk sterilization system.

**Figure 4 foods-12-00459-f004:**
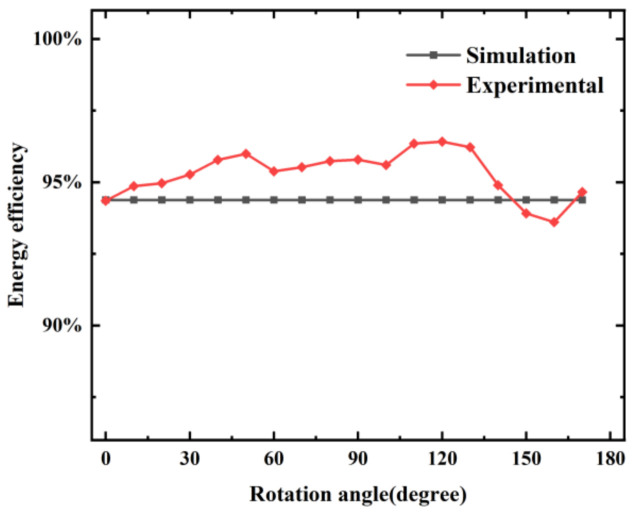
Simulation and experimental results of the S11 parameter when rotating the coaxial slot radiator.

**Figure 5 foods-12-00459-f005:**
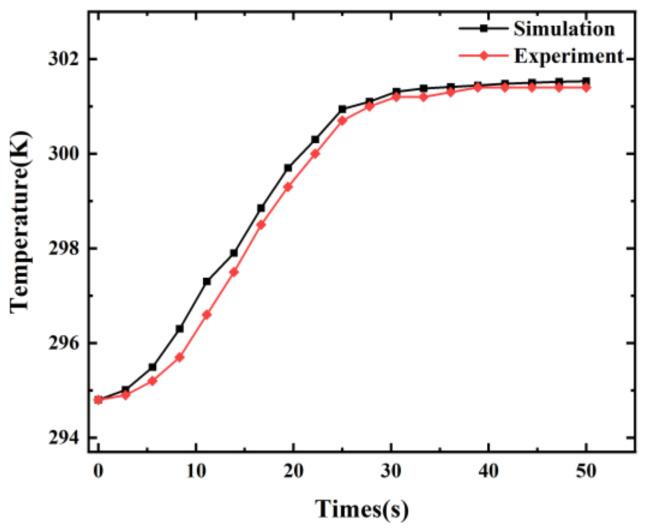
Simulation and experimental results on the average temperature of the outlet surface.

**Figure 6 foods-12-00459-f006:**
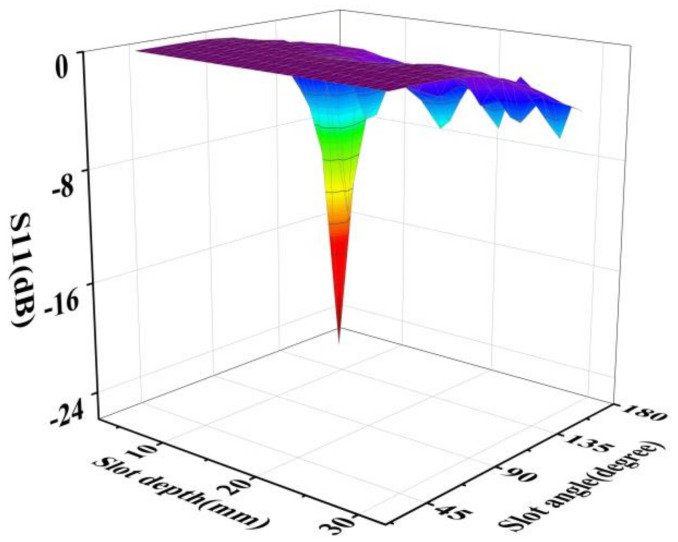
S11 simulation results of coaxial slot radiator for different values of slot depth and slot angle.

**Figure 7 foods-12-00459-f007:**
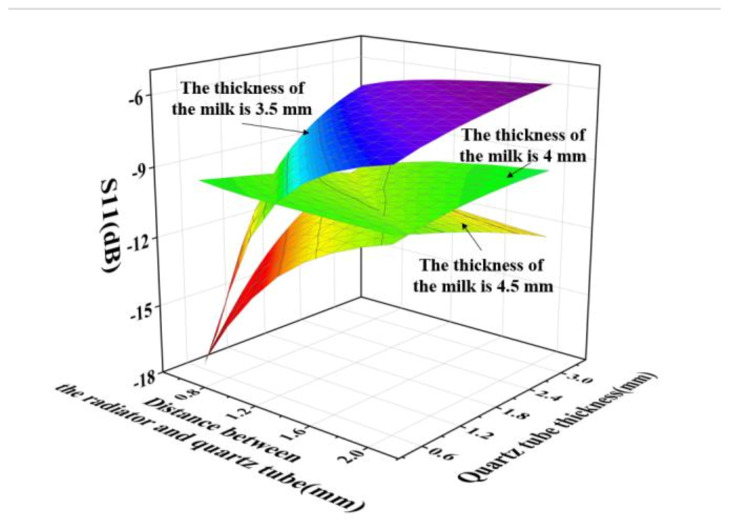
S11 simulation results of the microwave system for different dimensions.

**Figure 8 foods-12-00459-f008:**
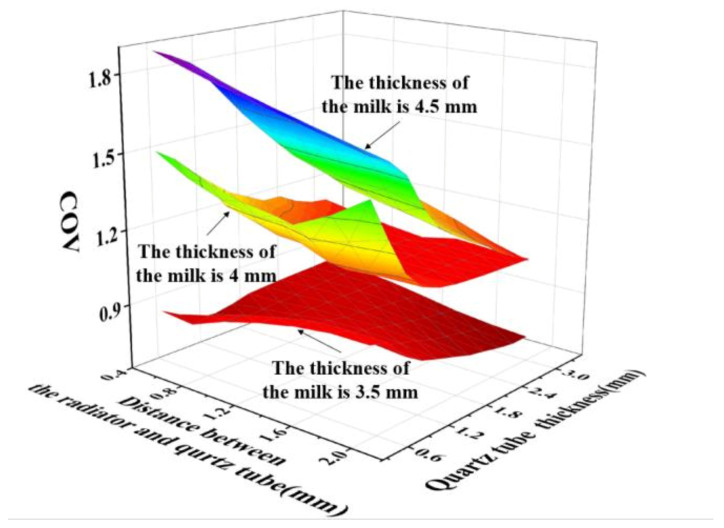
COV simulation results of the microwave system for different dimensions.

**Figure 9 foods-12-00459-f009:**
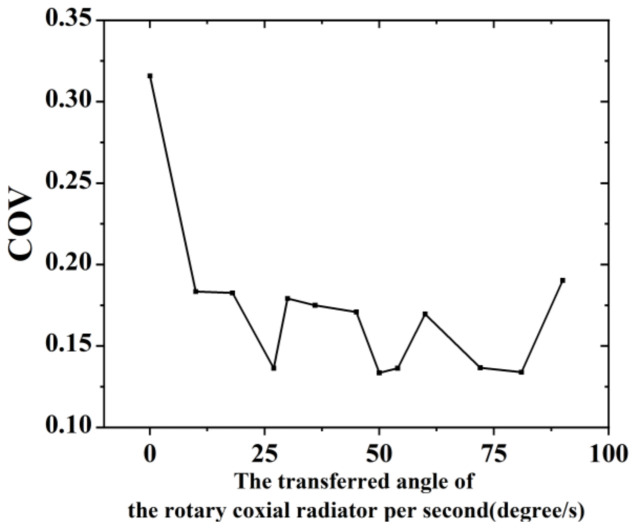
COV simulation results of the sterilization system for different rotation angle variations per second.

**Figure 10 foods-12-00459-f010:**
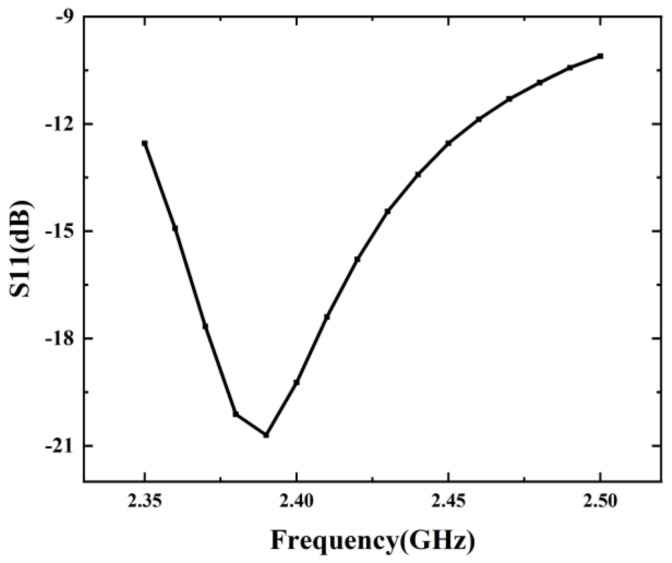
S11 values of the microwave continuous-flow sterilisation system at different frequencies.

**Figure 11 foods-12-00459-f011:**
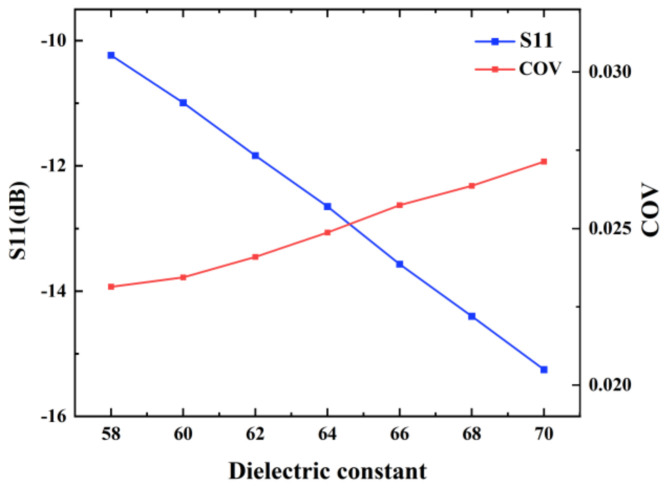
S11 and COV values of the microwave continuous-flow system for different dielectric constants of the liquid.

**Figure 12 foods-12-00459-f012:**
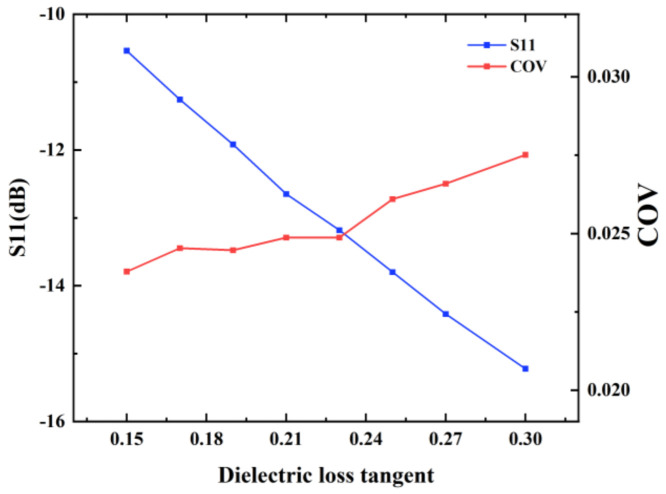
S11 and COV values of the microwave continuous-flow system for different dielectric loss tangents of the liquid.

**Figure 13 foods-12-00459-f013:**
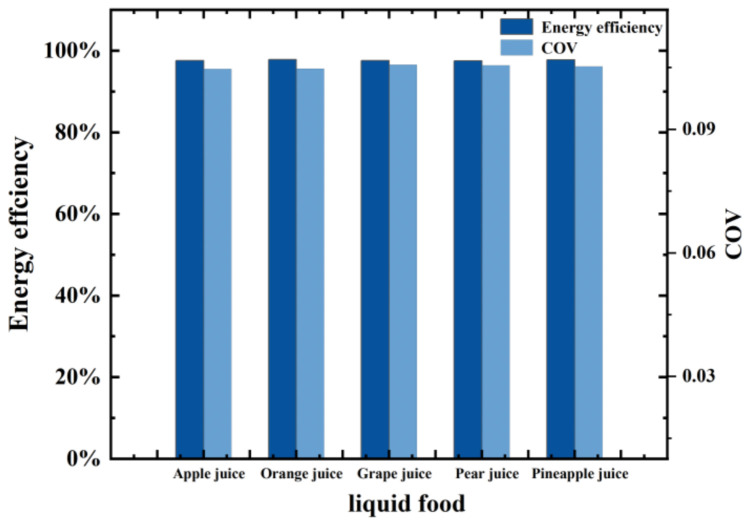
S11 and COV values of the continuous-flow sterilization system for different types of liquid food.

**Table 1 foods-12-00459-t001:** Related input parameters.

Property	Domain	Value	Unit
Relative permittivity	Air	1	--
	Milk	*ε* (T)	--
	Quartz	2.1	--
Relative permeability	Air	1	--
	Milk	1	--
	PTFE	1	--
	Aluminum	1	--
	Stainless steel	1	
Conductivity	Air	0	S/m
	Milk	0	S/m
	PTFE	0	S/m
	Aluminum	3.774 × 10^7^	S/m
	Stainless steel	1.3962 × 10^6^	S/m
Heat conductivity coefficient	Milk	*k* (T)	W/m·K
	Quartz	7.6	W/m·K
	Aluminum	238	W/m·K
	Stainless steel	*k′* (T)	W/m·K
Density	Milk	*ρ* (T)	kg/m^3^
Heat capacity at constant pressure	Milk	3935.6	J/kg·K
	Quartz	C_solid_1(T)	J/kg·K
	Aluminum	900	J/kg·K
	Stainless steel	C(T)	J/kg·K

**Table 2 foods-12-00459-t002:** The temperature-dependent dielectric properties of the milk (at 2.45 GHz).

T (K)	*ε_r_′*	*ε_r_″*
298.15	64.7083	13.0802
303.15	63.8003	13.1412
308.15	63.3613	12.4885
313.15	62.406	12.7017
318.15	61.856	12.6033
323.15	61.2887	12.3083
328.15	60.7587	12.2715
333.15	60.2187	12.2346
338.15	59.7797	12.1842
343.15	58.8244	11.2329
348.15	57.9164	11.1345

**Table 3 foods-12-00459-t003:** Comparison of simulation results of rotary coaxial slot antenna and stationary coaxial slot radiator heating.

	Average Export Surface Temperature	Export COV
Stationary coaxial slot radiator heating	346.97	0.3158
Rotary coaxial slot radiator heating (50°/s)	349.24	0.1335

**Table 4 foods-12-00459-t004:** Regression analysis of dielectric constants and dielectric loss factors of liquid food versus temperature T (°C) at 2.45 GHz.

Liquid Food	Dielectric Constant	Loss Factor
Grape juice	−0.221 T + 77.33	20.28 − 0.26 T + 1.94 × 10^−3^ T^2^ − 2.92 × 10^−5^ T^3^
Apple juice	−0.236 T + 80.51	19.06 − 0.32 T + 2.87 × 10^−3^ T^2^ − 8.17 × 10^−6^ T^3^
Pear juice	−0.258 T + 79.25	21.04 − 0.38 T + 3.62 × 10^−3^ T^2^ − 1.05 × 10^−5^ T^3^
Orange juice	−0.258 T + 81.78	21.69 − 0.37 T + 4.31 × 10^−3^ T^2^ − 1.73 × 10^−5^ T^3^
Pineapple juice	−0.343 T + 81.89	20.87 − 0.31 T + 3.25 × 10^−3^ T^2^ − 1.16 × 10^−5^ T^3^

## Data Availability

The data are contained in the article.

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
