# Peer review of "Continuous-Flow Microwave Milk Sterilisation System Based on a Coaxial Slot Radiator"

_foods, 2023, doi:10.3390/foods12030459_

Round 1

Reviewer 1 Report

General remarks

This work describes a microwave heat treatment device using a coaxial device in which the product to be heated is injected. The major drawback of this type of device is the impossibility of ensuring optimized adaptation of the device with respect to the load. Whatever the load, the dielectric properties (real and imaginary parts) change considerably with the temperature, causing a permanent loss of the device. This type of study is therefore essential to design an optimized device.

If the use of a coaxial exonerates the problems of possible eigenmode type solutions, the problem remains excessively complex since along the applicator the dielectric properties evolve spatially and temporally. The resolution of maxwell's equations seems excessive with respect to the treatment of a progation equation in cylindrical geometry but with spatial and temporal gradient of dielectric properties. The main symptom is thermal runaway specifically for milk.

The authors do not mention any of these issues in the context of their study. It would be desirable to explain the reasons for this oversimplification.

Perhaps this simplistic approach is common for the field of food applications (according to the bibliography provided). In the field of microwave heating, such an approach is totally dated and disconnected from real situations. In this simplistic context, one may be surprised at numerical data whose significant digits make no sense: p. 158, 160 (table 1), 273, 288, 334.

Specific remarks

For the design part of the applicator 3.2 which properties have been retained or is in the air? In this latter case, the study has no interest for the folowing! Fig. 9 describes the quality of adaptation of the device without this being mentioned or explained beforehand.

It would be interesting to make a simple calculation of average heating for the volume of applicator and the flow rate. The volume is close to 114 cm3 (radius 3.6 and 2.9) and Cp 3.9. For 100W we find 0.2 °K per second. Then 6°K for 30 seconds. According to Fig. 4 9°K for 30 seconds ! It would be useful to make some comments on this subject.

Reviewer 2 Report

This paper reports a simulation and practical demonstration of a microwave heating system equipped with a coaxial slot radiator for milk sterilization. The coaxial slot radiator was designed using FEM simulation, and the heating efficiency and uniformity were analyzed. This microwave system is small lab scale. How can this system be scaled up? Here are the comments to the authors.

Introduction

1) In the background section, the introduction of previous papers is redundant. It would be better to make it a little more compact and emphasize the novelty of the microwave device used in this paper.

2) Many Foods readers are not microwave specialists. The description of the previous microwave device configuration and the coaxial slot radiator in this paper is difficult to understand. It would be easier to understand if the differences were more clearly explained.

3) There are Sections 2 and 3, but Section 1 is not clearly indicated.

Multiphysics simulation

4) Table 1; Heat conductivity coefficient and heat capacity of microwave device should be considered (quartz, alumina, and stainless steel).

Results and Discussion

5) Figure 2; Please indicate a diagram of the microwave devices. The photograph is not enough to understand the whole system by the broad range of food engineers.

6) How was the temperature distribution measured in the practical system?

7) Table 3; Temperature-dependent dielectric properties of the juice should be considered.

8) Table 1, “Relative permittivity” and Table 3, “Complex relative permittivity” are not consistent.

Reviewer 3 Report

It is a very interested work, the study Continuous-flow microwave milk sterilisation system with internal radiation based on a coaxial slot radiator

1-change the title form of the article

2-Reformulate the summary

3-Add more reference to the introduction

4-The paper title must be changer and to be short for relcting the main edia of this paper.

5-The abstract must be precise and sumarize clearly the content of the paper.

6-The introduction must be enhanced by following those steps:

  -Present the motivation and context of the study.

  -Define the research problem and the field of application.

  -Add relevant case study with important results.

  -Add  short sentences that introduce your results.

7-A literature review section (related works) is strongly recommended to position the contribution in relation to the existing

8-The methodology must be clear and consistent.

9-Conclusion; The practical implications; Theoretical implications and limitations of the study should be clearly articulated.

Round 2

Reviewer 2 Report

I believe that the manuscript has been improved enough.